# AviTrap: A novel solution to achieve complete biotinylation

**Cheng Lu[1,2], Jack Bevers[2], Tulika Tyagi[2], Hao To[2], May Lin[1,2], Shu Ti[1], Gerry Nakamura[2], WeiYu Lin[2], Yongmei Chen[2], Yan Wu[2], Hong Li[1,2], Jiansheng Wu[3], Feng Wang[1,2]***

1 Department of Protein Sciences, Genentech Inc., South San Francisco, CA, United States of America,
2 Department of Antibody Engineering, Genentech Inc., South San Francisco, CA, United States of America,
3 Protein Sciences, Wuxi Biologics, San Francisco, CA, United States of America

* wang.feng@gene.com

## Abstract

Site specific biotinylation of AviTagged recombinant proteins using BirA enzyme is a widely used protein labeling technology. However, due to the incomplete biotinylation reactions and the lack of a purification method specific for the biotinylated proteins, it is challenging to purify the biotinylated sample when mixed with the non-biotinylated byproduct. Here, we have developed a monoclonal antibody that specifically recognizes the non-biotinylated Avi-Tag but not the biotinylated sequence. After a ten-minute incubation with the resin that is conjugated with the antibody, the non-biotinylated AviTagged protein is trapped on the resin while the fully biotinylated material freely passes through. Therefore, our AviTrap (anti-Avi-Tag antibody conjugated resin) provides an efficient solution for enriching biotinylated Avi-Tagged proteins via a simple one-step purification.

**Data Availability Statement:** All relevant data are within the paper.

**Funding:** The author(s) received no specific funding for this work.

## Introduction

Biotin is a water-soluble B vitamin that is ubiquitous in all living organisms. It is a crucial cofactor for carboxylase enzymes that facilitate the transfer of $CO_2$ groups during biosynthesis [1]. Biotin binds to the chicken egg white protein avidin and the homologous protein streptavidin with extremely high affinity (dissociation constant $K_D = 10^{-15}$ M) and specificity [2]. The interaction between biotin and avidin/streptavidin remains resilient even under denaturing conditions such as high temperatures, 6M guanidine hydrochloride, or 1% sodium dodecyl sulfate (SDS) [3, 4]. Additionally, the biotin molecule is very small in size (molar mass = 244.31 g/mol), making it one of the most commonly used protein modifications that can be employed in various applications such as protein immobilization, detection, and conjugation [5].

Protein biotinylation is the covalent attachment of biotin to a protein through either chemical or enzymatic reactions. Chemical biotinylation is nonspecific but versatile, targeting various functional groups including primary amines (on lysines and the N-terminus of a protein), carboxyls (on aspartic and glutamic acids, as well as the C-terminus of a protein), sulfhydryls (on cysteines), and carbohydrates (in glycoproteins). However, chemical biotinylation is inherently random, which leads to heterogeneous products. Additionally, the modification of

**Competing interests:** The authors have declared that no competing interests exist.

certain residues may interfere with protein activity or protein-protein interactions. On the other hand, site-specific biotinylation can be achieved enzymatically using the *Escherichia coli* (*E. coli*) biotin ligase BirA, which catalyzes the covalent linking of a single biotin to a specific lysine. Its natural substrate is the biotin carboxyl carrier protein (BCCP) subunit of the acetyl-CoA carboxylase [6]. Initially, a 75-amino-acid segment of BCCP was fused to a target protein for BirA biotinylation [7]. Through a combinatorial peptide library screening, an artificial 15-amino-acid peptide known as the AviTag (GLNDIFEAQKIEWHE) was identified as the minimum substrate for BirA [8]. The protein of interest can be cloned with an AviTag on either end, and subsequently biotinylated *in vivo* through co-expression with BirA and biotin in bacteria [9–11], yeast [12–14], insect [15], or mammalian cells [16, 17]. Alternatively, purified recombinant AviTag fusion proteins can undergo *in vitro* biotinylation on the central lysine residue in the AviTag by incubation with BirA, biotin, and ATP [5, 18, 19].

While the AviTag site-specific biotinylation system is convenient, it regrettably does not guarantee the production of fully biotinylated material. The efficiency of enzymatic biotinylation with BirA varies between 50–80% *in vivo*, or 80–100% *in vitro* [20]. Optimizing the reaction conditions or strategically placing the AviTag at different ends of the protein has been shown to improve the efficiency, but complete biotinylation is almost impossible to achieve. Therefore, an efficient approach to purify the biotinylated protein following the biotinylation reaction is highly desired. While avidin or streptavidin-coated resins or beads could be used to capture biotinylated proteins efficiently, the robust interaction makes protein elution impossible without using harsh and denaturing conditions [21, 22], which inevitably destroy the natural protein structure and compromise protein activity for subsequent biochemical applications.

In this work, we have generated an antibody specifically recognizing the non-biotinylated AviTag sequence, but not the same peptide with biotinylation. By conjugating the antibody to resins, we also developed a robust tool (named AviTrap) to trap the non-biotinylated species after the biotinylation reaction, so that only the biotinylated AviTagged protein is present in the flowthrough. This is a novel approach to enrich and purify active biotinylated proteins under very mild conditions.

## Materials and methods

### Generation of rabbit anti-AviTag peptide-specific antibodies

New Zealand White rabbits were immunized with AviTag peptide (GLNDIFEAQKIEWHE) coupled to keyhole limpet hemocyanin (KLH) or ovalbumin (OVA). All animal procedures were performed in accordance with the Animal Welfare Act. Ethical review of all animal activity was approved and performed by the contract research organization's Institutional Animal Care and Use Committee (Josman LLC IACUC), and all animal procedures were performed in accordance with Animal Care and Use Protocol (ACUP) that was approved by the IACUC (JLP-003.019). Appropriate anesthetics (isoflurane) and analgesics were used in study procedures when needed, as approved in the IACUC protocol. All animals were humanely euthanized using appropriate euthanasia agents (Euthasol/Beuthanasia) and methods approved in the IACUC protocol. After receipt of animals, they were acclimated for at least 48 hours prior to initiation of study. Rabbits were individually housed with food and water. Any materials in contact with the rabbits were of lab animal industry standards. All procedures listed in the protocol involving live animals (including euthanasia and anesthesia) were performed by trained, qualified personnel. The single peptide-specific IgG B cells were isolated using a modified protocol related to published literature [23]. This modified workflow included direct fluorescence-activated cell sorting (FACS) of IgG+ /peptide-specific B cells into single wells. The B

cell culture supernatants were assayed by enzyme-linked immunosorbent assay (ELISA) for binding to an AviTag peptide (CGLNDIFEAQKIEWHE, non-biotinylated) but not binding to the control peptide in which the lysine was replaced with an alanine (CGLNDIFEAQAIEWA). AviTag specific B cells were lysed and immediately frozen at -80°C for storage until molecular cloning. Variable regions (VH and VL) of each monoclonal antibody from the rabbit B cells were cloned into expression vectors from extracted mRNA as previously described [23]. Individual recombinant rabbit antibodies were expressed in Expi293 cells and subsequently purified with MabSelect SuRe antibody purification resin (Cytiva). Purified anti-AviTag antibodies were then subjected to specific biochemical binding assays and surface plasmon resonance (SPR) kinetic affinity screening.

## Engineering of anti-AviTag antibodies

Rabbit anti-AviTag antibodies were further engineered to increase binding to non-biotinylated AviTagged proteins by shuffling light and heavy chains for clones with high homology and using a Comprehensive Substitution for Multidimensional Optimization (COSMO) approach [24]. All residues in the complementarity-determining regions (CDRs) were substituted with different natural amino acids except cysteine and the wild-type amino acid. About 1200 variants were generated using PCR and then expressed transiently in Expi293 cells. Small scale antibody cultures were purified using MabSelect SuRe (Cytiva). Antibodies were screened by high throughput SPR for binding to biotinylated and non-biotinylated AviTagged proteins. Top variants were reformatted to THIOMAB [25] format with S400C mutated cysteine for resin generation.

## Recombinant protein expression and purification

Recombinant proteins were expressed in *E. coli* BL21(DE3) with either N-terminal or C-terminal His-Avi-tag and purified by Ni-NTA Agarose (Qiagen) followed by size exclusion chromatography on a Superdex 200 preparative SEC column (Cytiva), in 20 mM Tris, 150 mM sodium chloride (NaCl), 10% glycerol, 0.5 mM tris(2-carboxyethyl)phosphine (TCEP), pH 7.5. Biotinylated proteins were co-expressed with BirA in the presence of 50 μM D-biotin in *E. coli* BL21(DE3) and purified similarly.

## *In vitro* biotinylation of AviTag-fused proteins using BirA enzyme

AviTagged proteins in 50 mM Tris, 50 mM NaCl, pH 8.0 at 1–2 mg/ml were mixed with 10 mM magnesium acetate, 100 μM D-biotin, 10 mM ATP, 1:50 (w/w) FLAG-tagged BirA a, and incubated with gentle mixing on a rocking platform for 2 hours at 30°C, or overnight at 4°C. FLAG-tagged BirA was removed by anti-FLAG M2 Affinity Gel (Sigma-Aldrich), and the biotinylated proteins were subsequently purified by ion exchange chromatography or size exclusion chromatography.

## Antibody binding affinity analysis by SPR

Binding affinities of rabbit anti-AviTag antibodies were measured using a SPR Biacore-T200 instrument (Cytiva) at 25°C in HBS-P buffer (10 mM HEPES pH 7.4, 150 mM NaCl, 0.005% v/v Surfactant P20, Cytiva). Each anti-AviTag antibody at 0.3 μg/ml was immobilized on a different flow cell on a Series S sensor chip Protein A (Cytiva) with a flow rate of 10 μl/min for 1 min to achieve approximately 100 response units (RU). Four-fold serial dilutions of two different AviTag fusion proteins (N- and C-terminal tagged, 23.4 nM to 6000 nM) were injected at a flow rate of 30 μl/min for 120 sec followed by a 180 sec dissociation phase. The sensor chip was

regenerated by injection of 10 mM Glycine pH 1.5, at a flow rate of 30 μl/min for 30 sec. Sensorgrams obtained from the SPR measurements were analyzed by the double-subtraction method described by Myszka [26]. The signal from the reference flow cell was subtracted from the analyte binding response obtained from the flow cell with captured ligand. Buffer reference responses from multiple injections were averaged and then subtracted from analyte binding responses. The final double-referenced data were analyzed with Biacore T200 Evaluation software, version 3.1 (Cytiva). The equilibrium dissociation constant ($K_D$) was derived by fitting the equilibrium binding data to a 1:1 Langmuir binding model.

The further engineered anti-AviTag antibodies were characterized using a Biacore 8k+ or T200 instrument (Cytiva) at 25˚C in HBS-P buffer. Initial antibody binding was measured to biotinylated and non-biotinylated forms of AviTagged proteins using single-cycle kinetics. Top clones were screened in a multi-cycle format. Each antibody was captured on a Series S sensor chip Protein A (Cytiva) to ~6000 RU. Sensorgrams for binding of AviTagged proteins were recorded using an injection time of 120 sec at a flow rate of 30 μl/min. After injection, disassociation of the AviTagged protein from the antibody was monitored for a period of time in the running buffer. Double-referenced sensorgrams were analyzed using a 1:1 Langmuir binding model to calculate the kinetics and binding constants. Rmax values were normalized based on antibody capture levels.

### Antibody pull-down assay

To assemble the pull-down column, 10 μl of MabSelect SuRe resin (Cytiva) was transferred to an empty spin column and equilibrated with phosphate buffered saline (PBS). The resin was then mixed with 100 μg of each anti-AviTag antibody by gentle rocking at 4˚C for 10 min. Unbound antibodies were removed by PBS washes (4 x 300 μl). After that, 40 μg of artificially made 50% biotinylated protein by mixing biotinylated and non-biotinylated AviTag protein at 1:1 molar ratio was added to the antibody captured resin and mixed at 4˚C for 15 min. The column was then centrifuged at 4000 rpm and the flowthrough was collected and analyzed by Bradford protein assay and Liquid Chromatography Mass Spectrometry (LC-MS).

### Mass spectrometric analysis

LC-MS analysis was performed on a 6230 Time-of-Flight (TOF) LC-MS (Agilent). Samples were chromatographed on a PRLP-S column, 1,000 Å, 8 μm (50 mm × 2.1 mm, Agilent) heated to 80 ˚C. A linear gradient from 20–60% B in 4.3 min (solvent A, 0.05% trifluoroacetic acid in water; solvent B, 0.04% trifluoroacetic acid in acetonitrile) was used and the eluent was directly ionized using the electrospray source. Data were collected and deconvoluted using the Agilent Mass Hunter qualitative analysis software.

### Amine coupling of antibodies onto agarose resin

Anti-AviTag antibodies were buffer exchanged into Coupling Buffer (0.1 M sodium phosphate, 0.15 M NaCl, pH 7.2), and applied to Coupling Buffer-equilibrated AminoLink Coupling Resin (Thermo Scientific) at 20 mg/ml density. In a fume hood, Cyanoborohydride Solution (5 M sodium cyanoborohydride, $NaCNBH_3$ in 1 M sodium hydroxide, Thermo Scientific) was added to the reaction slurry to a final concentration of ~50 mM $NaCNBH_3$). The reaction was mixed by end-over-end rocking overnight at 4˚C. After removing the flowthrough, the resin was then washed with 2 resin-bed volumes of Coupling Buffer, and 2 resin-bed volumes of Quenching Buffer (1 M Tris, pH 7.4). The remaining active sites on the resin were then blocked by adding 1 resin-bed volume of Quenching Buffer with 50 mM $NaCNBH_3$ and mixed gently for 30 min by end-over-end rocking. The resin was then washed with at least

5 resin-bed volumes of Wash Solution (1 M NaCl). The final product was stored in PBS buffer containing 0.05% sodium azide.

### THIOMAB deblocking

The antibodies in THIOMAB format with S400C mutated cysteine were expressed in Expi293 or CHO cell lines, and purified by MabSelect SuRe (Cytiva) followed by size exclusion or SP cation exchange chromatography. The purified antibodies were adjusted to pH 8.0 using 1 M Tris pH 8.5, and supplemented with 2 mM ethylenediaminetetraacetic acid (EDTA), and 50-fold molar excess of dithiothreitol (DTT) to reduce the blocked cysteine residues. After LC-MS confirmation of complete blocker removal, the antibodies were diluted 10 times with Buffer A (10 mM Succinate, pH 5.0), and loaded onto a 5 ml HiTrap SP HP column. The column was then washed with 10 column volumes of Buffer A, and eluted with 50 mM Tris, 150 mM NaCl, pH 8.0. The purified reduced antibodies were then re-oxidized with 15-fold molar excess of dehydroascorbic acid (DHAA) and 2 mM EDTA. The samples were then loaded onto a 5 ml HiTrap SP HP column (Cytiva) and washed with 10 column volumes of Buffer A. The antibodies were then gradient eluted with 30–100% Buffer B (10 mM Succinate, 300mM NaCl, pH 5.0) in 20 column volumes.

### SulfoLink resin conjugation

A desired amount of SulfoLink Coupling resin (Thermo Scientific) was transferred to an empty column and equilibrated in Coupling Buffer (50 mM Tris, 5 mM EDTA-Na; pH 8.5) at room temperature. Anti-AviTag antibody was buffer-exchanged into Coupling Buffer and added to the column at 30 mg of antibody for 1 ml of resin. The column was mixed by rocking at room temperature for 15 min and then incubated upright at room temperature for an additional 30 min without mixing. The column was washed with three resin-bed volumes of Coupling Buffer. The nonspecific binding sites on resin were blocked by mixing with one resin-bed volume of 50 mM L-Cysteine hydrochloride (Thermo Scientific) in Coupling Buffer for 15 min at room temperature followed by incubation without mixing for an additional 30 min. The column was then washed with six resin-bed volumes of 1 M NaCl and two resin-bed volumes of PBS buffer and stored in PBS buffer with 0.05% sodium azide.

## Results

### Rabbit anti-AviTag antibody identification and characterization

The aim of this work was to develop an antibody that specifically recognizes the AviTag sequence (GLNDIFEAQKIEWHE) without biotin, but not the same sequence with biotinylation on the lysine residue (Fig 1). New Zealand White rabbits were immunized with the Avi-Tag peptide, and then the IgG+/AviTag peptide-specific B cells were directly sorted into single wells by FACS, which resulted in 2067 clones. The B cell culture supernatants were then screened by ELISA for AviTag peptide binders and 89 clones were identified. A second round of ELISA screening was performed using recombinant protein with an AviTag, followed by the counter-screening using biotinylated protein. There were 39 clones that showed specific binding to the non-biotinylated AviTag fusion protein. After molecular cloning of rabbit B cells and sequence analysis, 27 unique clones were identified and produced recombinantly for further characterization.

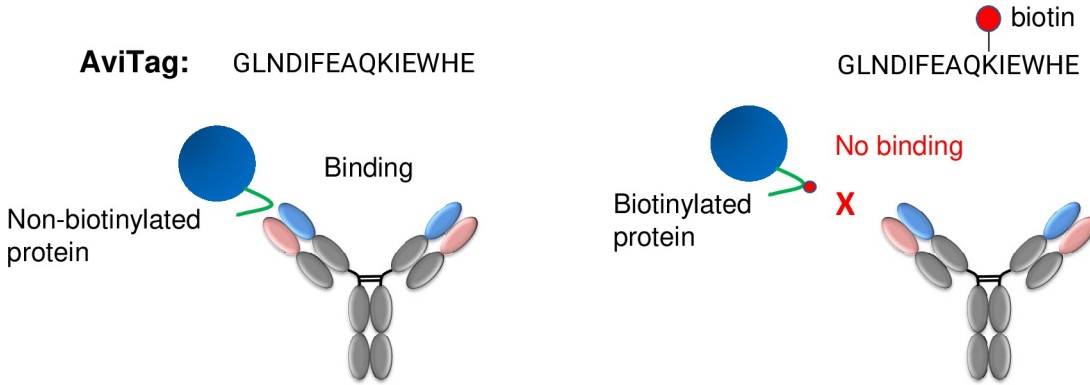

**Fig 1. Illustration of an antibody designed to recognize specifically the non-biotinylated but not the biotinylated AviTag.** A recombinant protein (blue sphere) with an AviTag (green tail) can be recognized by the antibody while the same fusion protein with a biotin (red dot) labeled on the lysine cannot.

### Binding affinity analysis by SPR

The 27 anti-AviTag antibodies identified from Rabbit B cells showed specific binding to the non-biotinylated AviTag by ELISA. Their binding affinities were determined by SPR against AviTag fusion proteins. A total of six different recombinant proteins with either an N-terminal or C-terminal AviTag were produced in *E. coli* and used throughout this study to assess the antibodies. Structurally, they are all in globular shape. We hereby referred to them as Proteins 1–6 and listed their molecular weights and AviTag positions in Table 1.

The antibodies were initially analyzed against Protein 1 with an N-terminal AviTag and the SPR results revealed binding with fast association and dissociation rates (Fig 2). The sensorgrams did not have sufficient curvature for kinetic measurement, hence the steady state affinity of each antibody was measured instead and listed in Table 2. Antibody 6F8 had the best binding affinity of 0.6 μM (Fig 2A), followed by 15A8 with a 0.8 μM. Another two antibodies, 6C4 (Fig 2B) and 2F8, showed about 1 μM affinities. Fifteen antibodies had binding affinities in the range between 2 μM—7 μM. The weakest antibodies, 22D12, 21D5, and 17D8, had affinities too low to be measured accurately, with the AviTag antigen protein titrated in the range from 23.4 nM to 6000 nM.

Next, we investigated whether the AviTag position would affect antibody binding. The same set of anti-AviTag antibodies were titrated with Protein 2 fused with a C-terminal AviTag. Their binding affinities were measured by SPR similarly and listed in Table 3. The affinity rankings of the 27 antibodies against the C-terminal AviTagged Protein 2 are quite similar compared to those against the N-terminal tagged Protein 1, although not in exactly the same order, and the top four best binders remain the same, with about 3 μM, 5 μM, 8 μM, and 10 μM affinities for 6F8 (Fig 2C), 6C4 (Fig 2D), 2F8, and 15A8, respectively. The binding

**Table 1. AviTag fusion proteins used in the study.**

| Protein | Molecular Weight (kDa) | AviTag Position |
|---------|------------------------|-----------------|
| 1 | 25 | N-terminus |
| 2 | 29 | C-terminus |
| 3 | 20 | C-terminus |
| 4 | 72 | N-terminus |
| 5 | 49 | N-terminus or C-terminus |
| 6 | 26 | N-terminus or C-terminus |

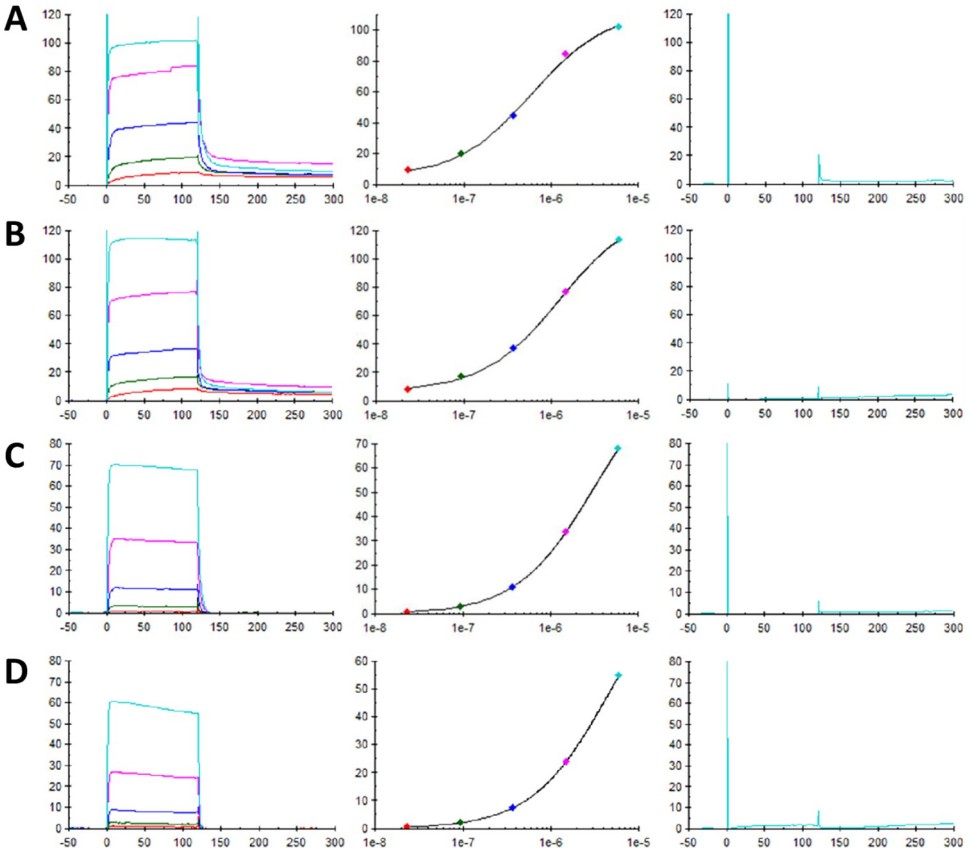

**Fig 2. SPR steady state affinity analysis.** SPR analyses of the rabbit anti-AviTag antibodies for their binding affinities towards AviTag fusion proteins. Two of the lead antibodies, 6F8 titrated with either N-terminal AviTagged Protein 1 (A) or C-terminal AviTagged Protein 2 (C), and 6C4 titrated with Protein 1 (B) or Protein 2 (D). The left panels are the SPR sensorgrams of the antibodies titrated with the non-biotinylated AviTag fusion proteins at 23.4 nM (red), 93.75 nM (green), 375 nM (blue), 1500 nM (magenta), and 6000 nM (cyan). The middle panels are their steady state affinity fittings. The right panels are the sensorgrams of the antibodies injected with the biotinylated AviTag fusion proteins at 6000 nM.

affinities of these antibodies against Protein 2 are in general about five times weaker, which may be due to the different nature of the target proteins, but the best ones are still in the single digit micromolar range, indicating their effectiveness despite where the AviTag was positioned in the protein.

To confirm the binding specificity of these anti-AviTag antibodies, we also checked the binding of each antibody to the two biotinylated AviTagged proteins by SPR. As we expected, no binding was detected in any of the anti-AviTag antibodies (Fig 2A–2D, right panels). Considering that these antibodies were raised with AviTag specific peptides and went through multiple rounds of screens and counter-screens with AviTag peptides and fusion proteins, we don't expect them to bind to the protein itself other than to the AviTag sequence. We tested several control proteins without an AviTag and no antibody binding was detected (data not shown).

## Anti-AviTag antibodies captured on Protein A resin can be used to remove non-biotinylated protein

As the binding affinities and specificities were assessed, we were interested in exploring if these anti-AviTag antibodies could be used in biotinylated protein purification. To test this, an

**Table 2. Steady state affinities of anti-AviTag antibodies against N-terminal AviTagged Protein 1 by SPR.**

| Antibodies | KD (μM) |
|---|---|
| 6F8 | 0.607 |
| 15A8 | 0.822 |
| 6C4 | 1.251 |
| 2F8 | 1.451 |
| 17B9 | 2.002 |
| 1A9 | 2.252 |
| 14G11 | 2.332 |
| 1G7 | 2.422 |
| 3E9 | 2.562 |
| 22A2 | 2.612 |
| 17E8 | 2.762 |
| 6D4 | 2.963 |
| 14C8 | 3.053 |
| 9E8 | 3.583 |
| 18A6 | 3.593 |
| 18B5 | 4.114 |
| 10D8 | 5.235 |
| 2B3 | 7.196 |
| 14D4 | 7.667 |
| 22C4 | 10.609 |
| 14B7 | 11.310 |
| 8B2 | 14.113 |
| 11G11 | 22.620 |
| 16H8 | 41.237 |
| 22D12 | ND |
| 21D5 | ND |
| 17D8 | ND |

antibody pull-down assay was designed to remove the non-biotinylated protein from a biotinylated protein mixture (Fig 3A).

Each rabbit antibody to be tested was incubated in a spin filter device with Protein A resin which captures the antibody by the Fc region. After washing away excess antibody, the anti-AviTag antibody-bound resin was incubated with a premade protein mixture containing 50% biotinylated and 50% non-biotinylated AviTagged protein. After centrifugation, the flowthrough was collected and analyzed by LC-MS for biotinylation level. When tested against the N-AviTagged Protein 1, about half of the antibodies could remove the non-biotinylated protein completely, or left only trace amounts that were barely detected (less than 1%). Fig 3B (left panels) shows the four lead antibodies (6F8, 6C4, 2F8, and 15A8) successfully removed the non-biotinylated portion of the protein mixture, as indicated by the left peak disappearance, leaving the biotinylated target protein (the right peak) in the flowthrough. On the contrary, the antibodies with weaker affinity, such as 10D8, 14C8, 14B7, and 11G11, left significant amounts of non-biotinylated protein in the flowthrough (Fig 3B, right panels). The protein concentration of the flowthrough was also checked to make sure that there was minimum nonspecific binding of biotinylated target protein to the resin. About 80% to 90% of the biotinylated material was recovered in each flowthrough, indicating no significant loss of the desired biotinylated protein.

**Table 3. Steady state affinities of anti-AviTag antibodies against C-terminal AviTagged Protein 2 by SPR.**

| Antibodies | KD (μM) |
|---|---|
| 6F8 | 3.103 |
| 6C4 | 4.514 |
| 2F8 | 7.997 |
| 15A8 | 9.719 |
| 1G7 | 12.411 |
| 1A9 | 12.511 |
| 17B9 | 14.012 |
| 6D4 | 14.813 |
| 22A2 | 15.614 |
| 3E9 | 15.914 |
| 17E8 | 27.024 |
| 17D8 | 27.024 |
| 2B3 | 28.425 |
| 22C4 | 28.625 |
| 10D8 | 36.733 |
| 8B2 | 48.143 |
| 14G11 | 72.965 |
| 14C8 | 102.091 |
| 9E8 | 153.136 |
| 18B5 | ND |
| 16H8 | ND |
| 14D4 | ND |
| 22D12 | ND |
| 18A6 | ND |
| 14B7 | ND |
| 11G11 | ND |
| 21D5 | ND |

The same experiments were repeated using the C-terminal AviTagged Protein 2 mixture. Similar results were obtained (Fig 3C), which confirmed the specificity and effectiveness of our lead antibody, as well as its potential to be developed into a purification tool.

## Affinity maturation of lead antibodies by CDR sequence engineering

Our pull-down assay results have proved the feasibility of developing an anti-nonbiotinylated-AviTag coupled resin as a purification tool. While it works with many recombinant protein formats, it is not compatible with antibodies, or Fc-tag fusion proteins, due to the fact that the Fc-fragment also binds to the Protein A resin. To circumvent this limitation, we decided to directly immobilize the anti-AviTag antibodies onto a solid support by covalent conjugation either randomly or site-specifically. To compensate for potential affinity loss during conjugation, the lead antibody candidates were further engineered to improve affinity and binding capacity.

Four lead antibodies, 2F8, 6C4, 6F8 and 15A8, with highly related CDR sequences in both the heavy and light chains, were selected based on SPR analysis and antibody pull down studies. Initially, heavy chain (HC) and light chain (LC) combinations of these four were expressed and tested for binding to AviTag fusion proteins. No significant increase was observed and the best combinations were 6C4 LC/6C4 HC and 6F8 LC/6C4 HC.

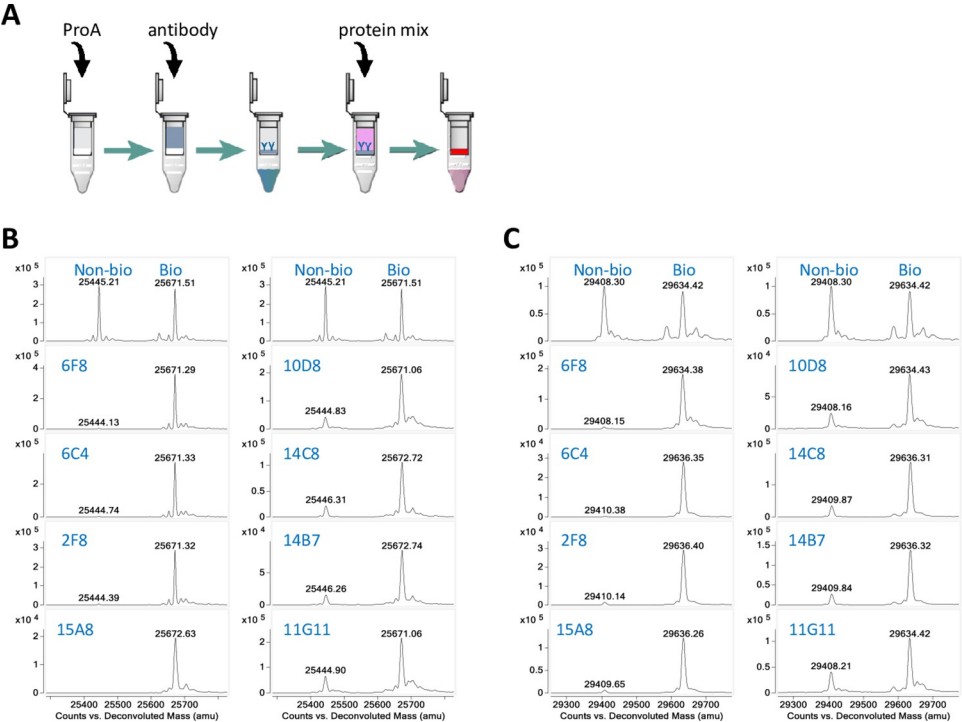

**Fig 3. Antibody pull-down assays confirmed binding. (A)** Schematic diagram of the antibody pull-down assay. Anti-AviTag antibody is added to a spin filter device with pre-applied Protein A resin. After the antibody is captured on the resin, the unbound antibody is spun down and washed away. A partially biotinylated AviTag fusion protein mixture is added to the antibody bound resin for incubation. After centrifugation, the non-biotinylated protein is expected to be trapped by the antibody on the resin, and the flowthrough is expected to contain fully biotinylated protein. **(B)** LC-MS deconvoluted spectra of an N-terminal AviTagged Protein 1 mixture before (top panels) and after incubation with different antibody-bound resin (lower panels labeled with antibody variants). The protein mixture has two main peaks corresponding to the non-biotinylated (~25.4 kDa) and the biotinylated Protein 1 (+226 Da, ~25.6 kDa) as marked. After passing through the resin capturing either antibody 6F8, 6C4, 2F8, or 15A8, only the biotinylated Protein 1 remained in the flowthrough. However, the weaker antibodies 10D8, 14C8, 14B7, and 11G11, could not remove the non-biotinylated protein completely. **(C)** LC-MS deconvoluted spectra of a C-terminal AviTagged Protein 2 mixture before (top panels) and after incubation with different antibody bound resin (lower panels labeled with antibody variants). The non-biotinylated (~29.4 kDa) and biotinylated (~29.6 kDa) peaks are marked. Lead antibodies 6F8, 6C4, 2F8, and 15A8 were able to remove most of the non-biotinylated Protein 2, leaving less than 1% in the flowthrough, while the weaker antibodies 10D8, 14C8, 14B7, and 11G11 left detectable amounts of non-biotinylated protein uncaptured.

Following that, saturation mutagenesis of the 6C4 LC and HC CDRs was performed using COSMO as previously described [24]. About 1200 variants were expressed transiently at small scale (1 ml) and screened using high-throughput SPR on a Biacore 8k+ system for binding to biotinylated and non-biotinylated AviTag fusion proteins. Variants were triaged based on improved binding to non-biotinylated protein while retaining no binding to biotinylated protein. Top 6C4 HC variants were also combined with 6F8 LC. Final lead variants (6F8 LC/6C4 HC.YW, 6F8 LC/6C4 HC.IF, 6F8 LC/6C4 HC.IL, 6F8 LC/6C4 HC.KM, and 6F8 LC/6C4 HC.KV) were expressed on large scale and then confirmed by binding to AviTag fusion proteins 1–3, on a Biacore T200. To mimic resin binding, these variants were captured at high density (~6000 RU) on a Protein A chip and single concentrations of non-biotinylated proteins were allowed to bind the captured antibodies. Rmax values normalized by capture levels showed increased binding for 6C4 variants to all three different recombinant proteins tested compared to the parental 6C4 antibody (Fig 4A).

To test whether these affinity matured antibodies could tolerate random conjugation, they were covalently coupled onto the AminoLink Coupling Resin through primary amines.

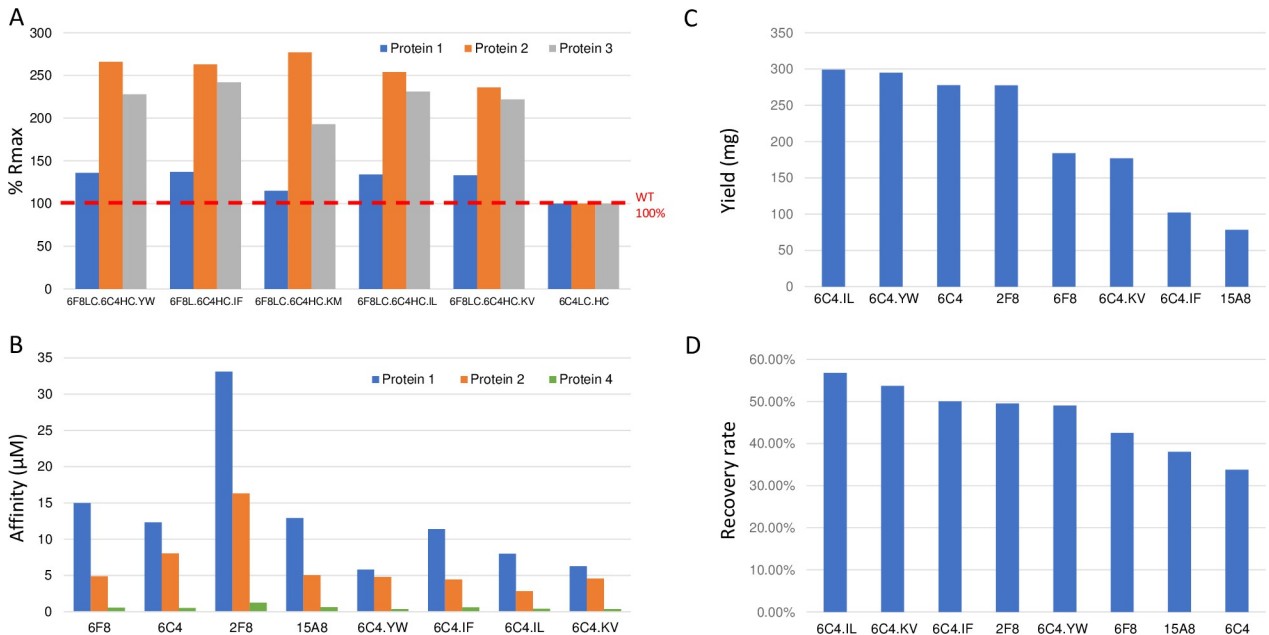

**Fig 4. Comparisons of mutagenesis-optimized antibodies. (A)** SPR normalized Rmax values from high capture, single injection experiments of 6C4 HC mutation variants against AviTag fusion proteins 1–3. Top five mutation variants were captured at about 6000 RU on a Protein A chip, and 1000 nM of non-biotinylated proteins were injected over the captured antibodies. Increased Rmax was observed for new variants compared to the original lead antibody 6C4, normalized as 100% by antibody capture level. **(B)** SPR steady state affinities of the four best mutagenesis-optimized antibodies as THIOMABs showed improvement compared to the four unoptimized lead antibodies against AviTag fusion proteins 1, 2, and 4. **(C)** Expression levels of the antibodies were compared in terms of yield from 1L of cell culture. **(D)** Recovery rates after deblocking of the THIOMABs were compared.

However, the random conjugation reaction killed the activity and none of them bound to the AviTagged proteins, likely due to the lysines in the CDRs being conjugated. To solve this problem, we decided to use the THIOMAB technology [25], which introduces an engineered cystine in the antibody that enables site-specific conjugation.

Four lead variants (6F8 LC/6C4 HC.YW, 6F8 LC/6C4 HC.IF, 6F8 LC/6C4 HC.IL, and 6F8 LC/6C4 HC.KV) together with the parental antibodies (6F8, 6C4, 2F8, and 15A8) were generated in THIOMAB format (S400C mutation was introduced in the HC) for comparison analysis. Biacore T200 measurement of the steady state binding affinities towards AviTag fusion proteins 1, 2, and 4, revealed that all four variants had relatively higher affinities than their parents, although the affinities were still in single digit micromolar range (Fig 4B). In consideration of large-scale antibody production for resin conjugation, the expression yield at 1L CHO cells of these antibodies was also compared as listed in Fig 4C. THIOMABs are expressed in mammalian cells with their engineered cysteines 'blocked' by glutathione or free cysteines, and only become reactive if 'deblocked' [27]. Therefore, we also compared the recovery rate after the deblocking process (Fig 4D). After THIOMAB cysteine reduction, re-oxidation, and purification, we were able to recover 57% of the IL, 54% of KV, 50% of YF, and 49% of YW mutant antibodies. Taken together, we selected 6F8 LC/6C4 HC.IL as the final lead anti-AviTag antibody.

## Antibody conjugated resin and purification method development

The THIOMAB 6F8 LC/6C4 HC.IL rabbit antibody was deblocked and covalently conjugated onto SulfoLink Coupling resin through the exposed sulfhydryls (-SH) on the engineered cysteine. The coupling density was between 20 to 30 mg of antibody per ml of resin.

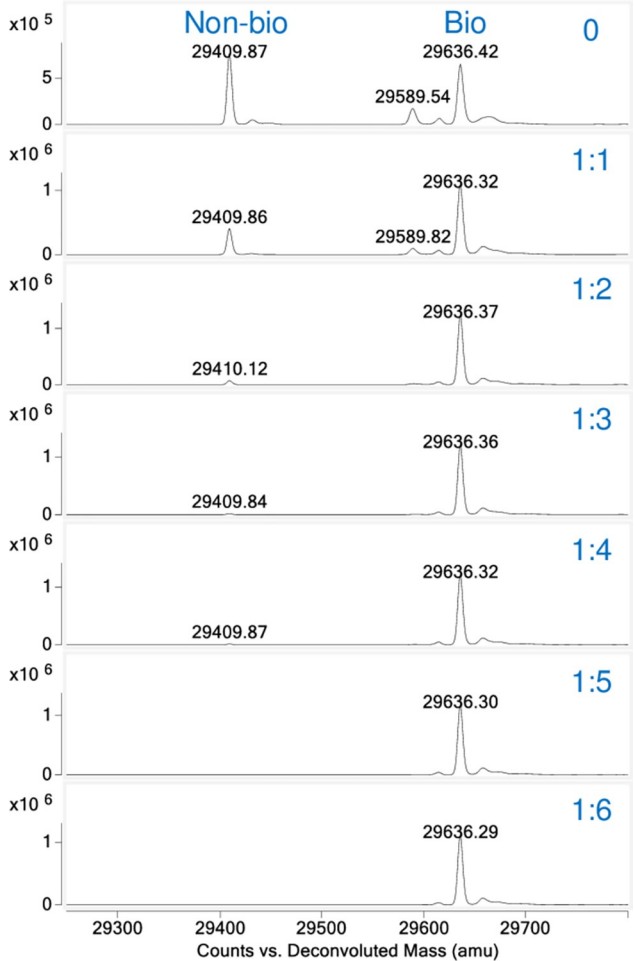

**Fig 5. Titration studies on the amount of anti-AviTag resin required to remove non-biotinylated protein completely by LC-MS.** Mass spec deconvoluted spectra showing a 50% biotinylated Protein 2 mixture treated with increasing amounts of anti-AviTag resin based on the molar ratio of non-biotinylated protein to monovalent antibody arms from 1:0 to 1:6 as labeled. The two major peaks are marked as non-biotinylated (~29.4 kDa) and biotinylated (~29.6 kDa). The non-biotinylated peak is shown to be completely removed with 1:5 resin and above.

To determine how much antibody-conjugated resin is needed for purifying biotinylated protein, a pull-down assay was performed using Protein 2 which showed lower binding affinity to the antibodies by SPR. The protein mixture (50% biotinylated with 50% non-biotinylated) was applied to the anti-AviTag antibody coupled resin in spin columns at different protein to antibody ratios. Since each antibody has two binding sites, the ratios in the pull-down assays were calculated as the non-biotinylated protein to single antibody arm at 1:1, 1:2, 1:3, 1:4, 1:5, and 1:6. The MS results clearly suggested that the flowthrough from the 1:1 experiment still had a notable amount of non-biotinylated protein that did not bind, while at a ratio of 1:3 or 1:4, the residual non-biotinylated left in the flowthrough was already hard to detect. At 1:5 ratio and above, the resin was able to capture the non-biotinylated protein completely (Fig 5).

We continued to determine the sensitivity of our antibody resin in differentiating the non-biotinylated protein from the biotinylated. A series of Protein 1 mixtures with different levels (10%, 50%, and 90%) of non-biotinylated "contaminants" were prepared and tested in the same pull-down assays, and the flowthroughs were analyzed by LC-MS. The results confirmed

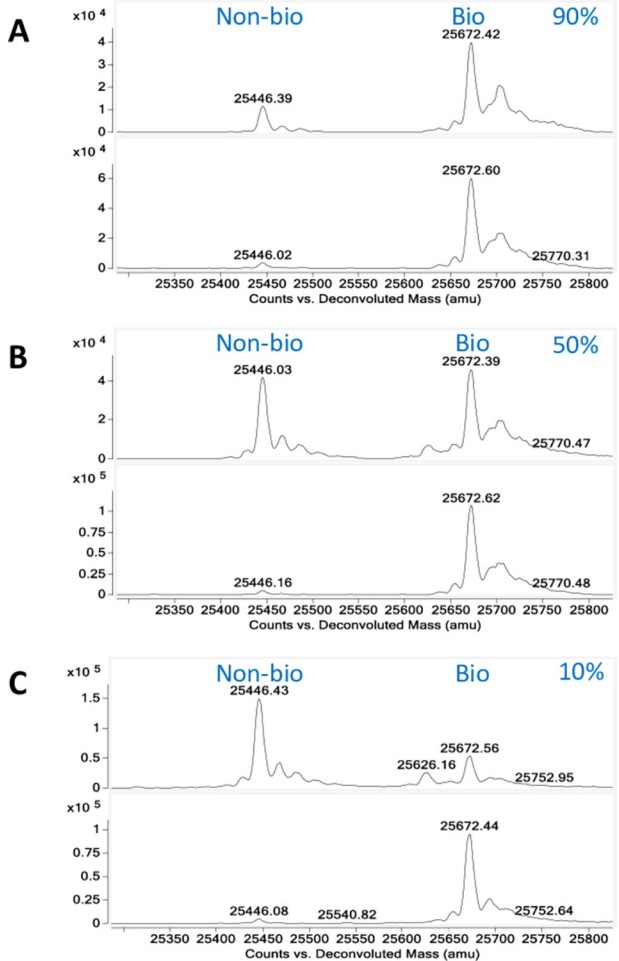

**Fig 6. LC-MS analysis of an AviTag fusion protein at different biotinylation levels treated with the anti-AviTag antibody resin.** The non-biotinylated Protein 1 (the ~25.4 kDa peak marked) was mixed with its biotinylated form (+226 Da, ~25.6 kDa peak marked) to obtain 90% (**A**), 50% (**B**), and 10% (**C**) biotinylation levels (top panels). They were purified by the anti-AviTag antibody resin to nearly 100% biotinylation (lower panels).

that all the samples were purified to above 99% biotinylation after incubating with the antibody resin (Fig 6), suggesting the resin binding is highly specific.

The aforementioned studies were carried out using different proteins (Proteins 1–4) with either an N- or C-terminal AviTag. Additionally, we produced two proteins (Protein 5 and 6), and each protein has two versions, with either an N- or C-terminal AviTag. To further validate the application in real-world scenarios, the Protein 5 pair were *in vitro* biotinylated with BirA to a level of about 55% and 45% biotinylation for the N- and C-AviTag constructs, respectively (Fig 7A and 7B, top panels). Following that, both samples were incubated with the anti-AviTag conjugated resin at the ratio of 1:6 (protein to antibody binding site), and the resulted flow-throughs both contained greater than 99% of biotinylated protein (Fig 7A and 7B, lower panels). Similar results were obtained with Protein 6. This set of constructs were co-expressed with BirA in *E. coli* which yielded ~50% and 70% biotinylation, respectively (Fig 7C and 7D, top panels). Once again, our resin effectively polished both samples (Fig 7C and 7D, lower panels).

Together, these results demonstrated the potential of our anti-AviTag antibody conjugated resin as a robust and versatile tool for polishing and enriching biotinylated AviTag fusion

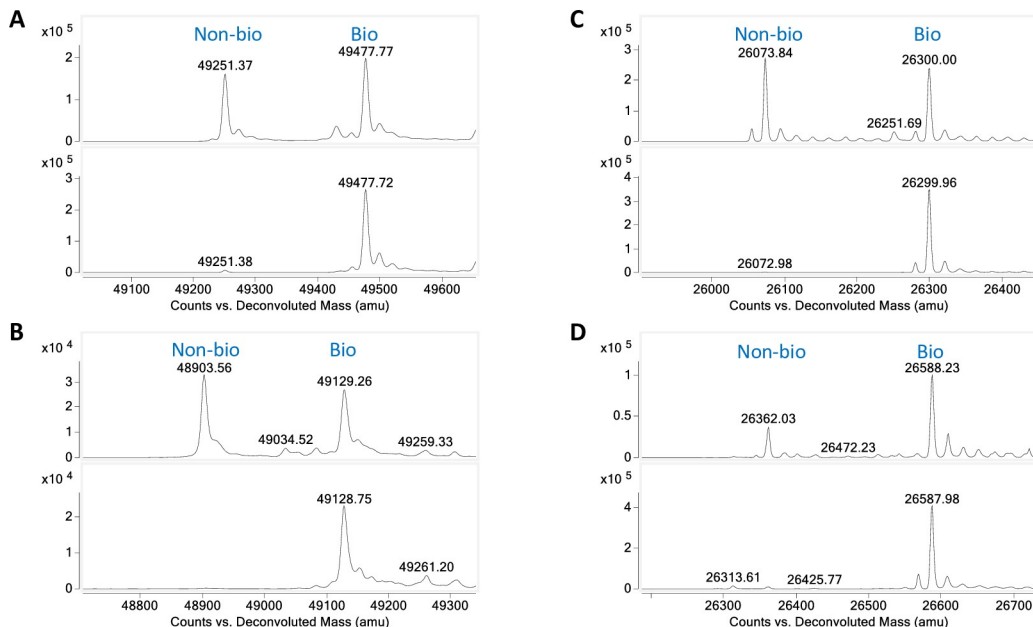

**Fig 7. Incomplete biotinylated proteins could be purified by the anti-AviTag antibody resin.** (**A**) LC-MS spectrum of the N-AviTagged Protein 5 *in vitro* biotinylated with BirA to ~55% (top panel). The two major peaks corresponding to the non-biotinylated and biotinylated proteins were labeled. After treatment with the anti-AviTag resin, the non-biotinylated peak was reduced to less than 1% (lower panel). (**B**) LC-MS spectrum showing the C-AviTagged Protein 5 *in vitro* biotinylated with BirA to ~45% (top panel). The two major peaks corresponding to the non-biotinylated and biotinylated proteins were labeled. After treatment with the anti-AviTag resin, the non-biotinylated peak was completely removed (lower panel). (**C**) LC-MS spectrum of the N-AviTagged Protein 6 biotinylated to ~50% by co-expression with BirA (top panel). The two major peaks corresponding to the non-biotinylated and biotinylated proteins were labeled. After treatment with the anti-AviTag resin, the non-biotinylated peak was completely removed (lower panel). (**D**) LC-MS spectrum showing the C-AviTagged Protein 6 biotinylated to ~70% by co-expression with BirA (top panel). The two major peaks corresponding to the non-biotinylated and biotinylated proteins were labeled. After treatment with the anti-AviTag resin, the non-biotinylated peak was reduced to less than 1% (lower panel).

proteins produced through *in vitro* and *in vivo* reactions. We hereby named the resin as "AviTrap".

## Discussion

The AviTag/BirA protein biotinylation system is one of the most widely used tools in biochemical and biomedical research due to the extremely high affinity and specificity of the interaction between biotin and avidin/streptavidin. However, the purification of functional biotinylated protein remains a challenge. As we mentioned earlier, avidin/streptavidin-coated resins or beads usually necessitates denaturing elution conditions, such as guanidine hydrochloride or SDS coupled with high heat [21, 22]. This may be tolerated for certain analysis, such as mass spectrometry and SDS-PAGE, but does not yield active protein for downstream applications. Several commercially available resins utilize chemically modified (strept)avidin molecules (e.g., nitrated avidin derivatives [28]), which exhibit reduced binding affinity for biotin and permit milder elution conditions at high pH as seen in CaptAvidin Agarose from Invitrogen. However, use of a high pH condition is still not ideal for most proteins maintain activity. Pierce Monomeric Avidin Agarose beads utilize monomeric avidin that has a lower affinity for biotin, and bound molecules could be eluted using free biotin [29]. Biotin Antibody Agarose beads from ImmuneChem Pharmaceuticals employ anti-biotin antibodies to capture biotinylated proteins and offer elution with free biotin at near-neutral conditions [30]. While

the latter two beads provide gentler elution conditions, they both require going through the full purification cycle including binding, washing, and elution steps. Additionally, extra steps such as desalting, dialysis, or gel filtration chromatography are needed to remove the free biotin used in competitive elution. In contrast, the AviTrap antibody resin described in this study is a single step polishing tool that captures and removes the non-biotinylated protein from the biotinylated protein without additional elution or purification steps.

Thanks to the well-designed immunization strategy and stringent screenings/counter-screenings, the anti-AviTag antibodies discovered in this study showed high specificity and selectivity, which can differentiate the subtle biotin addition on the short AviTag peptide. Through COSMO affinity maturation, the optimized anti-AviTag antibody bound to Avi-Tagged fusion proteins with sub-micromolar to single-digit micromolar binding affinity. By immobilizing this antibody onto resins, we invented a novel technology tool to polish the Avi-Tagged fusion protein after *in vitro* or *in vivo* biotinylation reactions. Upon a quick and simple incubation with the resin, the non-biotinylated AviTagged protein will be trapped in the resin, leaving only the biotinylated population enriched in the flowthrough. Therefore, we named our resin AviTrap.

We noticed that some AviTagged proteins showed very different expression levels or biotinylation potentials, depending on the position (N-terminus or C-terminus) of the AviTag. Nevertheless, AviTrap recognizes the non-biotinylated AviTag at either N- or C-terminus efficiently, with minimal interferences from target protein size or shape.

There are multiple ways to immobilize an antibody onto solid supports, such as affinity resin (Protein A, Ni-NTA resins) capture, random chemical conjugation (AminoLink Coupling Resin), and site-specific conjugation (SulfoLink Coupling Resin). We decided to choose the site-specific conjugation using THIOMAB technology for high consistency and it will allow applications for most circumstances, including for AviTagged proteins with Fc-fragments. The AviTrap antibody was also selected and optimized for excellent expression yield and conjugation recovery rate, which makes the production of AviTrap more cost effective. The current version of AviTrap is used in a disposable format, and regeneration conditions have not yet been explored. It may be possible to develop a reusable version of the AviTrap antibody by engineering these antibodies to release non-biotinylated protein in a pH dependent manner.

## Acknowledgments

We would like to thank Wei-Ching Liang for helping with SPR analysis, Dr. Dick Vandlen and Dr. Claudio Ciferri for advice and discussion, the Protein Production group and Guiping Du for helping with antibody purification.

## Author Contributions

**Conceptualization:** Shu Ti, Hong Li, Jiansheng Wu.

**Data curation:** Yongmei Chen, Hong Li.

**Investigation:** Cheng Lu, Jack Bevers, Tulika Tyagi, Hao To, May Lin, Gerry Nakamura, WeiYu Lin.

**Supervision:** Yan Wu, Jiansheng Wu, Feng Wang.

**Writing – original draft:** Cheng Lu, Feng Wang.

**Writing – review & editing:** Cheng Lu, Jack Bevers, Gerry Nakamura, Feng Wang.

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
