## [Decision Letter · Decision Letter 0]

10 Nov 2023

PONE-D-23-31657AviTrap: A novel solution to achieve complete biotinylationPLOS ONE

Dear Dr. Wang,

Thank you for submitting your manuscript to PLOS ONE. After careful consideration, we feel that it does not fully meet PLOS ONE’s publication criteria as it currently stands. Therefore, we invite you to submit a revised version of the manuscript that addresses the points raised during the review process.

We look forward to receiving your revised manuscript.

Kind regards,

Robert Chapman, Ph.D.

Academic Editor

PLOS ONE

Journal Requirements:

2.  Thank you for submitting the above manuscript to PLOS ONE. During our internal evaluation of the manuscript, we found significant text overlap between your submission and previous work in the [introduction, conclusion, etc.].

Please revise the manuscript to rephrase the duplicated text, cite your sources, and provide details as to how the current manuscript advances on previous work. Please note that further consideration is dependent on the submission of a manuscript that addresses these concerns about the overlap in text with published work.

[If the overlap is with the authors’ own works: Moreover, upon submission, authors must confirm that the manuscript, or any related manuscript, is not currently under consideration or accepted elsewhere. If related work has been submitted to PLOS ONE or elsewhere, authors must include a copy with the submitted article. Reviewers will be asked to comment on the overlap between related submissions (http://journals.plos.org/plosone/s/submission-guidelines#loc-related-manuscripts).]

We will carefully review your manuscript upon resubmission and further consideration of the manuscript is dependent on the text overlap being addressed in full. Please ensure that your revision is thorough as failure to address the concerns to our satisfaction may result in your submission not being considered further.

Additional Editor Comments:

This is a very nice paper - well executed and easy to read. NOTE: I acted as the second reviewer for this manuscript as it falls within my expertise, and conducted my review independently of the first reviewer. We both arrived at similar conclusions - without knowing the structure of the proteins you used to test the system the work is not reproducible (a key criterion for PLOSone). The lack of this also makes it hard to interpret the data and understand how well it would translate to other proteins. Please can you provide this information?

Reviewers' comments:

Reviewer's Responses to Questions

**Comments to the Author**

1. Is the manuscript technically sound, and do the data support the conclusions?

Reviewer #1: Partly

Reviewer #2: Yes

2. Has the statistical analysis been performed appropriately and rigorously? 

Reviewer #1: N/A

Reviewer #2: Yes

3. Have the authors made all data underlying the findings in their manuscript fully available?

Reviewer #1: Yes

Reviewer #2: No

4. Is the manuscript presented in an intelligible fashion and written in standard English?

Reviewer #1: Yes

Reviewer #2: Yes

5. Review Comments to the Author

Reviewer #1: In this paper, Lu et al develop a new process for purifiying biotinylated AviTag proteins from their non-biotinylated analogues. This is an important challenge, as many techniques rely on completely biotinylated proteins, yet AviTag biotinylation is often not complete, and while pull down assays of the biotinylated proteins with avidin/streptavidin are effective subsequent elution of the protein is difficult. The authors demonstrate that antibodies against the non-biotinylated AviTag can be used for affinity purification to address this challenge. This paper is well written and the experiments and results are of a high standard. I recommend acceptance of this work subject to the following minor revisions (in order of appearance):

i) Pg 4. “recognising only the non-biotinylated but not the biotinylated AviTag’: this is a bit confusing to reader at first, clearing up language would help understanding of the paper

ii) Pg 4. Please define KLH and OVA

iii) Throughout: Authors switch between SPR and Biacore which may be confusing to readers less familiar with SPR, stick to using this term.

iv) Pg 11. ‘Fifteen antibodies HAD binding affinities…’

v) Pg 13: The authors discuss using a ‘different recombinant protein with a C-terminal avitag’. This is my main comment on the paper – throughout, the authors never actually say what the proteins actually are, despite the fact that this will likely have a major effect on the effectiveness of their technology. This is a particularly prominent place in which this is problematic: is the ‘different protein’ the same parent protein but with C instead of N terminal AviTag, i.e. a different construct, or a completely different proteins? The implications of this are important. If it is a different protein entirely, then the authors cannot draw the conclusions about differences between N or C termini that they make on Page 13, the protein itself could be dictating how well the AviTrap system is functioning. Alternatively, if it is a different construct then their analysis is more warranted, but I believe it is still not valid since this has only been demonstrated on a single protein rather than being a generalisable feature. I appreciate that the nature of the proteins may be confidential, but in this case some vague description of the properties of the protein would be useful, e.g. are these 150 kDa antibodies they are purifiying, or 10 kDa.

vi) Pg 19, the demonstration that multiple resin types and conjugation methods are viable is nice, but it would be good if authors could expand a little on why this is needed vs. just using protein A columns. This only needs to be brief for benefit of non-specialist readers

vii) Pg 19. Recovery of antibodies after THIOMAB should be to 2 significant figures, not 3

viii) Pg 19. ‘Coupling density WAS between’

ix) Pg. 20 ‘the RESIDUAL non-biotinylated left….’

x) Pg. 20 Fig 5 caption (and some others) are not descriptive of the figure, they provide a summary of the data shown. This needs to be addressed to describe the data that is actually being shown, e.g. for Fig. 5 “Mass spec data showing….”

xi) Pg. 23 The language at the start of the paragraph ‘The successful discovery of..’ is too casual and should be made more scientific

xii) Pg. 23 and 24 there are a number of aspects that are discussed which are not included in the results or methods sections, e.g. the Fc-tag, the work with the Fab, the His-tag purification. These seem like valuable additions to the paper but also an afterthought in the way they are written. Methods for these are essential if they are to be included in the paper, but also some discussion beyond the vague description that is currently in the paper would be good. Equally, if these parts are removed from the paper, if the authors do not want to add this additional material this would not affect my positive assessment of the work and that it should be accepted with minor revisions.

Reviewer #2: This manuscript is reports a new method for the removal of non-biotinylated Avi-tagged proteins. The incorporation of the Avi-tag sequence (GLNDIFEAQKIEWHE) into proteins to direct the site specific biotinylation via the BirA enzyme is widely used. While quite efficient, the reaction does not always go to completion, leaving some non-biotinylated proteins. In this manuscript Wang et al report a new antibody which can recognise the Avi-tag sequence and remove the non-biotinylated proteins on an affinity column. This approach has not (to the best of my knowledge or searching), been reported before. Optimisation of the antibody is impressive, and the data is excellent. The authors go to the trouble of determining not just the Kds, but how much of their lead antibodies are needed to purify the protein mixtures. The system works very well - no binding is observed to the biotinylated sequences despite quite high affinities to the non-biotinylated versions.

A few minor questions which should be addressed before publication:

1) Kd for the 27 anti-avitag proteins is determined by SPR against “a recombinant protein” expressing the avi-tag sequence at the N- and at the C- terminus. Purification of the N-terminal protein using the 5 lead and 5 weakest antibodies is then determined using a protein-A pull down assay. What is the recombinant protein used? This is important - the work is not reproducible without this information. Later in figure 4 there are 2 more proteins - what are these? Without knowing the MW or the structure of the protein used here it’s hard to predict how well this would work in different sequences.

2) Can the authors comment on how selective the lead antibody is? We can see from the LCMS and SPR that the biotinylated versions do not bind, but how selective is it for the avi-tag sequence over other proteins?

6. PLOS authors have the option to publish the peer review history of their article (what does this mean?). If published, this will include your full peer review and any attached files.

Reviewer #1: **Yes: **Chris Spicer

Reviewer #2: No

---

## [Author Response · Author response to Decision Letter 0]

22 Dec 2023

Dear Dr. Chapman,

I hope this letter finds you well. I am writing in response to the comments provided by the reviewers for our manuscript entitled, “AviTrap: A novel solution to achieve complete biotinylation”, which was submitted to PLOS ONE. We greatly appreciate the constructive feedback and suggestions from the reviewers.

We have carefully addressed each of the reviewers’ comments and made revisions to enhance the clarity and overall quality of the manuscript. It is with great pleasure that we are resubmitting our manuscript for further consideration. We hope that our edits and the responses provided below have successfully addressed all the review comments.

To facilitate your review, the following is a point-by-point response to the comments delivered in your letter dated 11/10/2023. The location of each revision was listed based on the marked-up manuscript showing all revisions inline. 

1. We ensured that our manuscript meets the PLOS ONE’S style requirements, including those for file naming.

2. To address the text overlap issue, we rewrote the introduction and discussion sections.

3. We reviewed the reference list and ensured that it is complete and correct.

Reviewer #1 Comments:

i) Pg 4. “recognising only the non-biotinylated but not the biotinylated AviTag’: this is a bit confusing to reader at first, clearing up language would help understanding of the paper

• Response: Thank you for this suggestion. We cleared up the language in Pg 4, lines 85-86.

ii) Pg 4. Please define KLH and OVA

• Response: Thank you for pointing this out. We have incorporated the definitions for both abbreviations in Pg 5, line 95. 

iii) Throughout: Authors switch between SPR and Biacore which may be confusing to readers less familiar with SPR, stick to using this term.

• Response: Thank you for this suggestion. We made changes throughout the manuscript and sticked to the term SPR.

iv) Pg 11. ‘Fifteen antibodies HAD binding affinities…’

• Response: Thank you for the correction. We incorporated this change in Pg 14, line 293.

v) Pg 13: The authors discuss using a ‘different recombinant protein with a C-terminal avitag’. This is my main comment on the paper – throughout, the authors never actually say what the proteins actually are, despite the fact that this will likely have a major effect on the effectiveness of their technology. This is a particularly prominent place in which this is problematic: is the ‘different protein’ the same parent protein but with C instead of N terminal AviTag, i.e. a different construct, or a completely different proteins? The implications of this are important. If it is a different protein entirely, then the authors cannot draw the conclusions about differences between N or C termini that they make on Page 13, the protein itself could be dictating how well the AviTrap system is functioning. Alternatively, if it is a different construct then their analysis is more warranted, but I believe it is still not valid since this has only been demonstrated on a single protein rather than being a generalisable feature. I appreciate that the nature of the proteins may be confidential, but in this case some vague description of the properties of the protein would be useful, e.g. are these 150 kDa antibodies they are purifiying, or 10 kDa.

• Response: You raised an important question. The proteins we originally used were two different proteins. To address your concern, we tested our lead antibody on two new sets of protein constructs with either an N- or C-terminal AviTag on the same protein. We added these experiments in Pg 25, lines 483-499, and replaced the previous Fig 7 with new data. We also included the description of the 6 proteins used in this study, in Pg 14, lines 280-283, and Table 1. 

vi) Pg 19, the demonstration that multiple resin types and conjugation methods are viable is nice, but it would be good if authors could expand a little on why this is needed vs. just using protein A columns. This only needs to be brief for benefit of non-specialist readers

• Response: Thank you for your suggestion. We included the explanation in the result section, Pg 20, lines 385-389.

vii) Pg 19. Recovery of antibodies after THIOMAB should be to 2 significant figures, not 3

• Response: Thank you for pointing this out. We corrected the significant figures accordingly in Pg 23, lines 441-442.

viii) Pg 19. ‘Coupling density WAS between’

• Response: Thank you for the editing suggestion. We made this change in Pg 23, line 448.

ix) Pg. 20 ‘the RESIDUAL non-biotinylated left….’

• Response: We appreciate very much for your careful editing. We made the correction in Pg 24, line 458.

x) Pg. 20 Fig 5 caption (and some others) are not descriptive of the figure, they provide a summary of the data shown. This needs to be addressed to describe the data that is actually being shown, e.g. for Fig. 5 “Mass spec data showing….”

• Response: Thank you for your comments. We revised captions for Figs 2-7.

xi) Pg. 23 The language at the start of the paragraph ‘The successful discovery of..’ is too casual and should be made more scientific

• Response: Thank you for this comment. We rewrote the whole discussion section including this paragraph in Pg 28. 

xii) Pg. 23 and 24 there are a number of aspects that are discussed which are not included in the results or methods sections, e.g. the Fc-tag, the work with the Fab, the His-tag purification. These seem like valuable additions to the paper but also an afterthought in the way they are written. Methods for these are essential if they are to be included in the paper, but also some discussion beyond the vague description that is currently in the paper would be good. Equally, if these parts are removed from the paper, if the authors do not want to add this additional material this would not affect my positive assessment of the work and that it should be accepted with minor revisions.

• Response: Thank you for your assessment. While these experiments expanded the AviTrap applications, we agree with you that they may cause confusion to the readers. We removed them from Pg 29. 

Reviewer #2 Comments:

1) Kd for the 27 anti-avitag proteins is determined by SPR against “a recombinant protein” expressing the avi-tag sequence at the N- and at the C- terminus. Purification of the N-terminal protein using the 5 lead and 5 weakest antibodies is then determined using a protein-A pull down assay. What is the recombinant protein used? This is important - the work is not reproducible without this information. Later in figure 4 there are 2 more proteins - what are these? Without knowing the MW or the structure of the protein used here it’s hard to predict how well this would work in different sequences.

• Response: Thank you for your comments. You raised an important question. As we mentioned above in our response to the reviewer #1 comment v), we incorporated descriptions of the properties, including the MW and structure shape, of the proteins we used, in Pg 14, lines 280-283, and Table 1.

2) Can the authors comment on how selective the lead antibody is? We can see from the LCMS and SPR that the biotinylated versions do not bind, but how selective is it for the avi-tag sequence over other proteins?

• Response: Thank you for this suggestion. We have elaborated on the selectivity discussions in Pg 18, lines 328-332, that the antibodies do not bind to control proteins without the AviTag sequence. We did test the lead antibody to two new sets of proteins (protein 5 and 6), and each protein has two versions: either N- or C-terminal AviTagged. The lead antibody showed the same selectivity for the AviTag sequence, in Pg 25, lines 483-499, and in Fig. 7. Therefore, we concluded that our lead antibody is very selective for the AviTag sequence. 

Additionally, we revised the materials and methods section by adding methods for recombinant protein expression and purification in Pg 6, in vitro biotinylation of AviTag-fused proteins using BirA in Pg 6-7, and mass spectrometric analysis in Pg 9. We also merged the “Characterization of anti-AviTag engineered antibodies by SPR” in Pg 10-11 with “Antibody binding affinity analysis by SPR” in Pg 7-8. With new data added into the manuscript, we updated and rearranged the authorship list, in Pg 1, lines 6-7. 

Concluding remarks:

Thank you very much for giving us the opportunity to strengthen our manuscript with your valuable comments and suggestions. We have addressed all the comments and incorporated all your feedback into our manuscript, with new data added. We look forward to hearing from you and hope the revised manuscript will be accepted at PLOS ONE. 

Merry Christmas and Happy New Year!

Sincerely,

Feng Wang, Ph.D.

Director/Senior Principal Scientist

Protein Sciences Department

Genentech Inc.

---

## [Editor Report · Decision Letter 1]

28 Dec 2023

AviTrap: A novel solution to achieve complete biotinylation

PONE-D-23-31657R1

Dear Dr. Wang,

We’re pleased to inform you that your manuscript has been judged scientifically suitable for publication and will be formally accepted for publication once it meets all outstanding technical requirements.

Kind regards,

Robert Chapman, Ph.D.

Academic Editor

PLOS ONE

Additional Editor Comments (optional):

All the comments have been addressed. Although the exact sequence of the protein is not given, the molecular weights and structural details are now provided